# DiSciPLE: Learning Interpretable Programs for Scientific Discovery

## Abstract

Creating hypotheses for new observations is a key step in the scientific process of understanding a problem in any domain. A good hypothesis that is interpretable, reliable (good at predicting unseen observations), and data-efficient; is useful for scientists aiming to make novel discoveries. This paper introduces an automatic way of learning such interpretable and reliable hypotheses in a data-efficient manner. We propose DiSciPLE (Discovering Scientific Programs using LLMs and Evolution) an evolutionary algorithm that leverages common sense and prior knowledge of large language models (LLMs) to create hypotheses as Python programs for scientific problems. Additionally, we propose two improvements: a program critic and a program simplifier to further improve our method to produce good hypotheses. We evaluate our method on four different real-world tasks in two scientific domains and show significantly better results. For example, we can learn programs with 35% lower error than the closest non-interpretable baseline for population density estimation.

## 1 Introduction

The scientific process involves the collection of data, the creation of hypotheses to explain the data, and experimental testing and validation of the hypothesis to produce scientific laws. This process is manual, laborious and time consuming. With advances in LLMs, there is an opportunity to automate the scientific pipeline to greatly accelerate scientific discovery and progress. While prior work in automating science has looked at efficient data collection (Van Horn et al., 2015; Sener & Savarese, 2018), the possibility of automatic hypothesis generation from observational data has been less explored. This is especially important because in many domains like earth science, there are vast troves of data waiting to be analyzed. A method that could automatically scour through this data to identify hypotheses for scientists to explore can significantly speed up the scientists' workflow. Our goal is to address this gap and produce an automatic hypothesis generation framework (fig. 1).

What are good desiderata for an automatic hypotheses generation framework? First, clearly the generated hypothesis should fit the observations well. This requires that the framework should be able to search through a large and *expressive* search space. Second, the generated hypotheses should be *reliable*, in that it should correctly predict data even outside of the observations used to generate the hypotheses. Third, a good hypothesis is one that is *interpretable* by scientists, since only then can scientists understand what is being discovered, and frame experiments to test the hypothesis. Finally, the hypothesis generation framework should be sample *efficient*. In other words, scientists coming up with a hypothesis should not need to collect a lot of observations to come up with a good

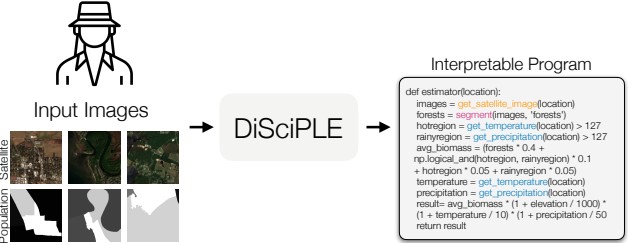

Figure 1: Our method asks an expert for a problem of interest and the observation data for that problem. Using this information it comes up with interpretable programs as hypothesis explaining the observation data.

hypothesis. In this paper, we ask: *How can we automatically discover expressive, interpretable and reliable hypotheses in a sample-efficient way?*

Existing approaches that discover hypotheses or formulae are not up to the task. Interpretable ML techniques such as concept-bottlenecks (Koh et al., 2020) can produce simple bag-of-words programs for visual classification but are not expressive enough to produce more sophisticated hypotheses. At the other end of the spectrum, black-box neural networks are expressive but offer no insight to the scientists for exploration. Symbolic regression (Cranmer, 2023) techniques are interpretable and can produce sophisticated formulae, but are very slow to converge because of the large search space. This issue is particularly severe in scientific domains where the hypothesis may reference any concept or variable in the scientific vocabulary, making the search space effectively unbounded. Neurosymbolic program learning (Johnson et al., 2017) address this search space with reinforcement learning, but this requires exceptional amount of training data and is not sample-efficient.

To be able to search through a large and expressive search space, recent works have looked at leveraging the common sense abilities of large language models (LLMs) (Surís et al., 2023). However, while zero-shot generation from LLMs can work well for well-known problems, they are insufficient for scientific domains with novel problems and data. Recent work has looked at leveraging LLMs as the mutation operator in evolutionary algorithms (Romera-Paredes et al., 2024; Chiquier et al., 2024). We leverage the common sense and prior knowledge of LLMs in tandem with an evolutionary algorithm-based symbolic regression method to generate programs as hypotheses, resulting in a significantly smaller search space. The interpretable programs are learned on a set of lower-level primitives, including neural modules such as open-world segmentation. Our framework produces neuro-symbolic programs as good hypotheses.

While performing crossover and mutation with LLMs can produce meaningful programs, several improvements can be made on the framework to further improve the search. First, unlike symbolic regression, the hypotheses evaluation in the evolution loop does not have to be limited to a single regression score. The LLM can leverage more information about the current hypothesis to better guide the search. Therefore, we propose to add a *critic* to our framework that can provide fine-grained information about the hypothesis to better guide the search. Our critic divides the training observation set into fine-grained categories and can provide feedback on categories that the hypothesis does well on or does badly on. This results in programs that can have an all-round better performance. Second, while the LLM is good at exploring the search space, many programs produced by the framework contain frivolous variables, that may not aid the hypotheses. Having such redundant features/parts in programs can result in more complex final hypotheses as well as a larger search space during evolution. Therefore, we also add an analytical (non-llm-based) program simplification method, that can remove such frivolous parts of the program.

We run experiments in two different expert domains (demography and climate science) on four real-world problems and show that we can produce good hypotheses by leveraging LLMs. In all of these cases, the programs generated are *reliable* and result in better generalization than deep neural nets. Moreover, in a few cases, we also observe better performance on even unseen in-distribution data.

Our contributions are:

- We introduce a novel framework **DiSciPLE** (**Di**scovering **Sci**entific **P**rograms using **L**LMs and **E**volution), that can produce interpretable, reliable, and sample-efficient hypotheses for diverse scientific applications.

- We present two key components: a critic and a program simplification method to DiSci-PLE that can further improve the evolutionary search resulting in better hypotheses.

- We show application of DiSciPLE on real-world problems by applying it on two scientific domains and four different problems and show that our learned hypotheses are indeed more interpretable, reliable, and data-efficient compared to baselines.

## 2 RELATED WORKS

**Concept bottlenecks.** Concept bottleneck (Koh et al., 2020) is an approach used to create interpretable-yet-powerful classifiers. The key idea is to train a deep model to predict a set of low-level concepts or bottlenecks and then learn a linear classifier. Such concept bottlenecks have

the basis of methods in several areas such as fine-grained recognition (Ferrari & Zisserman, 2007; Huang et al., 2016; Zhou et al., 2018; Tang et al., 2020) and zero-shot learning (Lampert et al., 2013; Akata et al., 2015; Kodirov et al., 2017). However in order to train these models, expensive data is needed to be collected for the bottleneck concepts themselves. One way to reduce this annotation cost is to sequentially ask question in an information theoretically optimized way (Chattopadhyay et al., 2024a;b). Researchers have also automated this pipeline by using large-language models as a knowledge base to propose concept bottleneck models (Menon & Vondrick, 2023; Pratt et al., 2023; Han et al., 2023). Chiquier et al. (2024) proposed an evolutionary algorithm with LLMs as the mutation operation to discover interpretable concept bottleneck models without prior information. While these models are interpretable, they are very simple in terms of expressive power. In this work, we instead evolve programs, which are more expressive than bag of words, while being interpretable.

**Symbolic regression.** Symbolic regression (Cranmer, 2023) is a powerful technique for learning such programs on evolution through evolutionary search. Several methods have been proposed to improve the search (Makke & Chawla, 2024), however most symbolic regression techniques cannot solve problems beyond simple mathematical formula. This is partly because the search space of possible hypotheses is too large due to combinatorial explosion. Like ours, recent work in symbolic regression (Grayeli et al., 2024) have also looked at using LLMs to better guide the search. However, they also only test this method on mathematical formula, with limited set of primitives. In real-world problems, the primitives functions are more complex than mathematical operations and can even be open-world for example a text-to-image segmentor.

**Neuro-Symbolic Program Learning.** (Mao et al., 2019; Dong et al.) is another avenue for learning programs as hypotheses for observation datasets or answering questions. These methods typically try to learn both discrete program structures together with neural networks. However, since this optimization is non-differentiable these methods require reinforcement learning (Johnson et al., 2017) or complex non-differentiable optimization techniques (Ellis et al., 2021). The hard optimization issue makes the problem of learning programs sample inefficient in real-world settings. We alternatively use LLMs ability to program to better guide the search for such programs.

**Program synthesis with LLMs.** Several works have utilized LLM coding ability in different applications such as VQA (Surís et al., 2023; Gupta & Kembhavi, 2022) and robot manipulation (Liang et al., 2023). While the zero-shot inferred code work very well on domains well-known to the internet, they tend to perform poorly on problems in scientific domains, as shown by our results.

**Scientific applications.** Researchers in numerous scientific domains have looked machine learning tools to build predictive models/hypotheses for their quantities of interest. In this work, we experiment in two such scientific domain of: demography and climate Science. In demography, we focus on the problems of socioeconomic indicator prediction (Yong & Zhou, 2024), namely population density and poverty estimation (Metzger et al., 2024; Xie, 2017). Similarly in climate science we focus on two problems of aboveground biomass prediction (AGB) (Nathaniel et al., 2023) and Contiguous Solar Induced Chlorophyll Fluorescence(CSIF) forecasting (Zhang et al., 2018).

## 3 METHODOLOGY

Our key contribution is a hypothesis generation framework that leverages LLMs and performs evolutionary search. In section 3.1 we formalize the problem of hypothesis generation. In section 3.2 we present our method of incorporating LLMs in the evolutionary search framework. Finally, In section 3.3, 3.4 and 3.5, we discuss the improvements to this framework to speed-up the search.

### 3.1 PROBLEM FORMULATION

**Data/Observation:** To come up with hypotheses, scientists first collect data/observations for a problem they are trying to understand. We represent a collection of $n$ such observations using $\mathcal{D} = \{(x_1, y_1), (x_2, y_2), \ldots, (x_n, y_n)\}$. Here, $x_i \in X$ is the input data/covariates to the hypothesis, and $y_i \in Y$ is the true observed output for a quantity of interest. For the problem of "population density estimation from satellite images", $X$ could be the set of satellite images for different regions

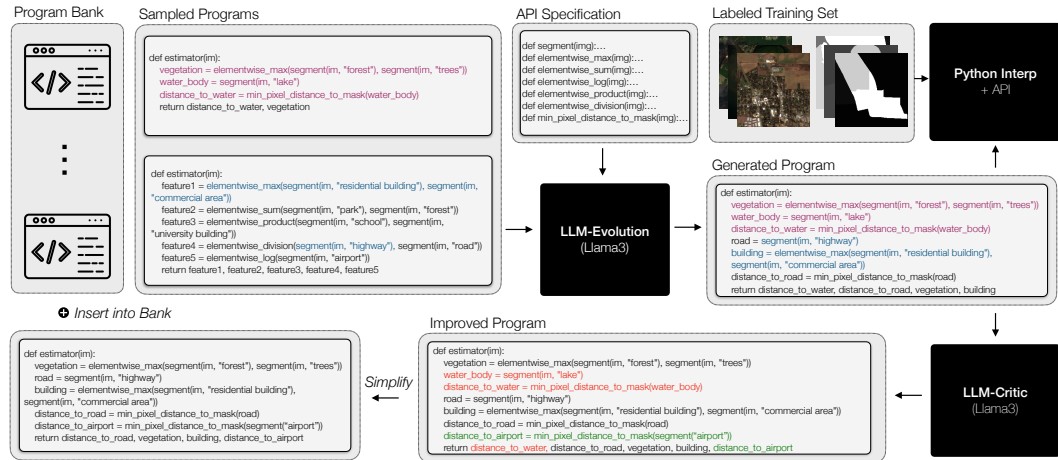

Figure 2: Overview of our evolutionary algorithm with *critic* and *simplification*. We start with an initialized bank of program trying to solve a task. From this bank we sample pairs of programs based on their fitness score and perform crossover/mutations over them to produce new programs. The generated program is further improved by passing it through a critic and then an analytical simplification step. This program is then evaluated and put in the next generation of program bank. The evaluation score of the program is used to determine the fitness for the next iteration of evolution.

and $Y$ would be population density maps for the corresponding satellite images collected through census. Our goal is to discover an interpretable hypothesis $h$ that can explain observations $\mathcal{D}$. Since we want our hypothesis to be data-efficient, $n$ is typically small: in the order of a few thousand.

**Objective:** The scientist or expert also provides a natural language name or description *descr* of the quantity of interest $Y$. For example, this may be the phrase "population density". As we will see below, this information will be useful in guiding the LLM to search the space well.

**Metric:** The scientists or domain experts also provide a hypothesis evaluation metric $\mathcal{M}$. A good hypothesis can lead to the best evaluation score

$$s(h; \mathcal{D}) = \frac{1}{n} \sum_{i=1}^{n} \mathcal{M}(h(x_i), y_i) \tag{1}$$

For example, in the case of population density a good metric used by domain experts is L2 error over log *i.e.* $\mathcal{M}(y', y) = ||log(y') - log(y)||_2$ (Metzger et al., 2022; 2024).

**Primitives:** Finally, the experts also provide our framework with a set of primitive variables and functions $\mathcal{F} = \{f_1, f_2 \ldots f_k\}$, that can be used to construct a hypothesis $h$. This set of primitive variables can even be unbounded or open-vocabulary. For example, for scientists analyzing remote-sensing image data, these primitives can include an open-vocabulary segmentation model (Mall et al., 2023). The primitives also include mathematical, logical, and image operations that can be composed to create powerful hypotheses. Examples of such primitives could be logarithm operation, elementwise maximum, or a distance transform respectively. Finally, the primitives can also be used to query specific attributes of the data, such as the average temperature or average precipitation at a location. The API of primitive functions can also change depending on the application domain. Please refer to section 4.2 for primitive API specifications for individual problems.

## 3.2 EVOLUTIONARY SEARCH

In evolutionary program search, we typically start with a large population of random programs. These programs are then sampled based on their fitness as parents. The parent programs create new programs through crossover and mutation, resulting in a new population. Newer generations improve over the previous as the population is getting optimized for the fitness function. The metric $\mathcal{M}$ can be used as the fitness function in our work.

---

**Algorithm 1:** DiSciPLE's learning loop

---

**Input:** Observation set $\mathcal{D} = \{(x_1, y_1), (x_2, y_2), \ldots, (x_n, y_n)\}$, metric $\mathcal{M}$, an objective prompt $p_o$, a set of primitive function $\mathcal{F} = \{f_1, f_2 \ldots f_k\}$

**Hyperparameters:** Mutation probability $\rho_m$, total number of generations $T$, population size $M$, crossover prompt $p_c$, and mutation prompts $p_m$.

**Output:** A hypothesis $h^*$ in the form of a program that explains the observations best.

1 $H^0 \leftarrow \{\}$ // Initialize a population of programs
2 **for** $i = 1, \ldots, M$ **do**
3   $\quad h_i^0 \leftarrow \mathcal{LLM}(p_o)$
4   $\quad H^0 \leftarrow H^0 \cup \{h_i^0\}$
5 $h^* \leftarrow h_i^0$
   // Learn a $\mathcal{D}$ that discriminates the training set $\{(x, y)\}$
6 **for** $t = 0, \ldots, T$ **do**
7   $\quad H^{t+1} \leftarrow \{\}$
8   $\quad$ **for** $i = 1, \ldots, M$ **do**
9     $\quad\quad h_{k_1}^t, h_{k_2}^t \leftarrow \text{sample\_parents}(H_t)$ // Sample parents for crossover
10    $\quad\quad h_i^{t+1} \leftarrow \mathcal{LLM}(h_{k_1}^t, h_{k_2}^t, s(h_{k_1}^t; \mathcal{D}), s(h_{k_2}^t; \mathcal{D}), p_o, p_c)$ // crossover operation
11    $\quad\quad$ **if** $u \sim \mathcal{U}(0, 1) < \rho_m$ **then**
12      $\quad\quad\quad h_i^{t+1} \leftarrow \mathcal{LLM}(h_k^{t+1}, s(h_k^{t+1}; \mathcal{D}), p_o, p_m)$ // mutation operation
          $\quad\quad$ // critique and simplification
13    $\quad\quad h_i^{t+1} \leftarrow \text{critic}(h_i^{t+1})$ $h_i^{t+1} \leftarrow \text{simplifier}(h_i^{t+1})$
14    $\quad\quad$ **if** $s(h_i^{t+1}; \mathcal{D}) > s(h^*; \mathcal{D})$ **then**
15      $\quad\quad\quad h^* \leftarrow h_i^{t+1}$
16    $\quad\quad H^{t+1} \leftarrow H^{t+1} \cup \{h_i^{t+1}\}$
17 **return** $h^*$;

---

In our method, we keep the overall algorithm the same but replace key steps in using LLMs. First, at the start of the process, we provide the LLM with a prompt for the objective to generate the initial programs. To leverage the prior knowledge of LLMs, we use a prompt $p_o$ that mentions the expert specified description of the quantity of interest: "*Given a satellite image, write a function to estimate $\langle descr \rangle$*". While an LLM cannot answer such a difficult scientific question without leveraging the observations $\mathcal{D}$, the prompt prevents the evolutionary algorithm from searching in completely random directions. As a result, our initial population is not entirely random.

Second, rather than using the symbolic methods of crossover and mutation, we use LLM to perform these operations. LLMs have common sense about programming and result in much better program modifications when performing crossover and mutations. More specifically, let $h_{k_1}^t$ and $h_{k_2}^t$ be two programs sampled from the $t^{th}$ generation selected as parents based on the fitness function. To perform a crossover operation we pass, the objective prompt $p_o$, the two programs $h_{k_1}^t$ and $h_{k_2}^t$, their corresponding scores (using eq. (1)), along with a crossover prompt $p_c$ to obtain a new program:

$$h_k^{t+1} = \mathcal{LLM}(h_{k_1}^t, h_{k_2}^t, s(h_{k_1}^t; \mathcal{D}), s(h_{k_2}^t; \mathcal{D}), p_o, p_c) \tag{2}$$

The crossover prompt instructs the LLM to make use of the two-parent program and come up with a new program. The LLM is able to combine elements from the parents to produce something new as can be seen in fig. 2. Please refer to the appendix C.1 for more examples.

Similar to crossover, we also mutate a program with some probability using a mutation prompt $p_m$.

$$^m h_k^{t+1} = \mathcal{LLM}(h_k^{t+1}, s(h_k^{t+1}; \mathcal{D}), p_o, p_m) \tag{3}$$

We present the exact input fed to the LLM for crossover and mutation in the appendix A. Note that both crossover and mutation operations also include the objective prompt preventing the LLM from generating programs far away from the objective.

### 3.3 FEATURE SET PREDICTION

Hypotheses for many problems require combination of multiple feature. Optimizing for the correct combination of these features is challenging for an LLM, as we are not doing gradient-based optimization. Therefore instead of prompting the LLM to directly generate a predictor for $y$, we prompt is to create a list of predictive features. We can learn a linear regressor on top of this list and use the regressor with the list of features as hypotheses. Both the program and regressor are interpretable, therefore our hypothesis is also interpretable.

### 3.4 PROGRAM CRITIC

The only form of supervision our method gets is through the metric score $s(h; \mathcal{D})$ created by evaluating the program $h$ on the observations. However, since we perform crossover and mutation through a language model, we can provide finer-grained information to LLM in order to aid the search.

More specifically, we propose a critic that performs a finer-grained evaluation of the program that we get after crossover/mutation. Our critic performs a *stratified evaluation* by partitioning the observation data into multiple categories and evaluating the program on individual strata.

$$\mathcal{D} = d_1 \cup d_2 \cup \cdots \cup d_c, \quad \text{where } d_i \cap d_j = \emptyset \text{ for } i \neq j \tag{4}$$

In all the applications where we use satellite images, we partition the observation dataset into land-use categories, using an open-world segmentation model. The critic obtains per-partition score $s(h; d_i)$ and prompts the LLM to improve the program on categories the model is bad. The addition of a critic improves the programs on data overlooked by programs, resulting in reliable programs.

### 3.5 PROGRAM SIMPLIFICATION

Successive steps of crossovers and mutations of programs result in large programs with many redundancies, hurting interpretability. We propose an analytical approach to simplify the programs and remove the redundant parts of it. Our hypothesis are programs without loop, they can be represented as a directed acyclic graphs (DAG) (we use the abstract syntax tree (AST); see fig. 5). In these DAGs, all the constants and the arguments of the function are root nodes. Only the return statement and the unused variables are the leaf nodes. Any leaf node that is not a return statement, is a piece of code that is not needed and can be removed. We then recursively remove all the leaf nodes that are not return statements.

While removing such nodes (unreachable by the return statement) is useful, there could still be features that are returned by the program but are not contributing. Recall that we use linear regression on the list of features returned by the program. The weights assigned to individual features by the regression model are useful indicators of which features are redundant. We remove the features that have a significantly smaller weight compared to the largest weight in the regression (a threshold of 5% works well). Removing these features from the return statement results in numerous newly created leaf nodes. We, therefore, redo the recursive leaf node removal to further simplify the program. Each program generated by mutations and crossover is first improved through the critic and then simplified before adding back to the population. Algorithm 1 shows the complete algorithm.

## 4 RESULTS

### 4.1 IMPLEMENTATION DETAILS

We used the open-source LLM *llama-3-8b-instruct* (Dubey et al., 2024) served using the vLLM library (Kwon et al., 2023). A majority of our evaluation tasks use satellite images, so to allow inferring semantic information from it, we use a black-box open-world foundational model for satellite images, GRAFT (Mall et al., 2023). Some experiments use ground-truth annotations from Open-StreetMaps (Vargas-Munoz et al., 2020) as an alternative to disentangle the effect of segmentation.

We run our evolutionary method for $T = 15$ generations with a population size of $M = 100$. For all the problems, the input observation data comes from different geographical locations around the world. We split this data into three parts. Two-thirds of the easternmost observations are used

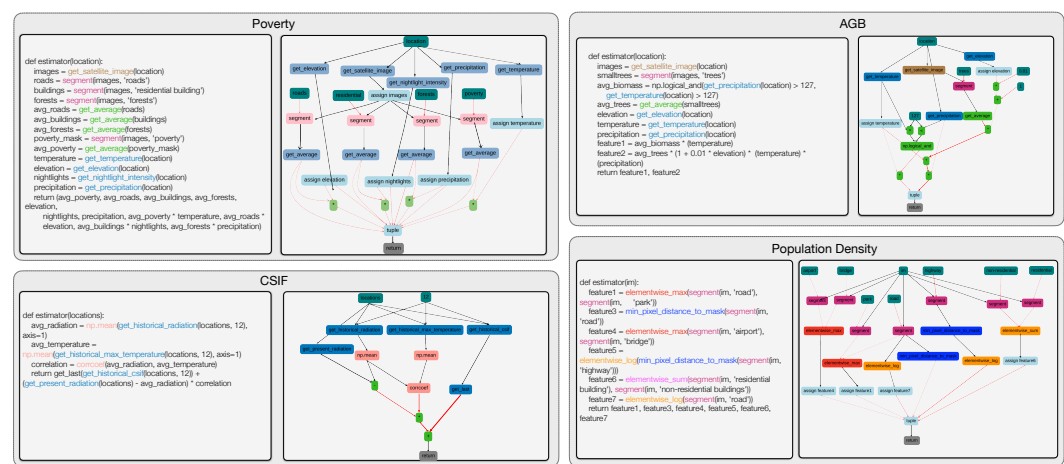

Figure 3: The best performing hypotheses for each of the 4 problems as Python programs (left in each card) and the corresponding DAG representation on the right. The DAG representation allows better visualization of the importance of different components. The thickness of the red edges determine how important that component is. A black edge represents computation; when removed it is either the same as one of its subsequent edges or removing it could result in a bug.

to create a training-testing split. The remaining one-third of the data is use to evaluate reliability (out-of-distribution generalization). We will release all four datasets for future research in this area.

## 4.2 SCIENTIFIC DOMAINS AND PROBLEMS

We apply DiSciPLE to two different problems in *Demography*: population density and poverty prediction. and to two problems in *Climate Science*: for AGB estimation and CSIF forecasting. In the following subsections, we present the observation datasets, metrics, and overview of primitives.

### 4.2.1 POPULATION DENSITY

**Observation Dataset**: The problem seeks to predict the population density by observing the satellite images of a region (Metzger et al., 2022; 2024). We obtain the population density values ($y_i$) for various locations in the USA by using ACS Community Surveys 5-year estimates (United States Census Bureau, 2024). Input observations ($x_i$) are sentinel-2 satellite images at a resolution of 10m (Drusch et al., 2012). For this experiment, we also use OpenStreetMaps masks (Vargas-Munoz et al., 2020) for 42 different land-use concepts (see appendix F.1) as part of the input.

**Metric and Primitives:** Population density values are aggregated at the county block group level. The predicted population densities are therefore also aggregated at the county block group level. The metric is the per-block group level average L2 error after applying a log transformation. Along with the arithmetic, and logical primitives (appendix B), we use open-vocabulary segmentation as a primitive. The segmentation function returns a binary mask for an input concept.

### 4.2.2 POVERTY INDICATOR

**Observation Dataset:** For poverty estimation, we use data from SustainBench (Yeh et al., 2021). The dataset contains coordinate location as input and wealth asset index as output.

**Metric and Primitives:** We use L2 error for each location as the evaluation metric. To obtain semantic land use information about a location, we first define a *get_satellite_image* function, that returns a sentinel-2 satellite image for any location. This can be used in conjunction with the open-world satellite image recognition model to obtain semantic information about the world. Other than this we also include as primitives functions that return average annual temperature, precipitation, nightlight intensity, and elevation at the input location.

Table 1: Performance of our hypotheses on unseen in-distribution observations across various problems. In most cases, DiSciPLE produces better hypotheses (**red** is best and *blue* is second best).

| | Spatial | | | | | | Temporal | |
| | Population Density | | Poverty | | AGB | | CSIF | |
| | L2-Log | L1-Log | L1 | RMSE | L1 | RMSE | L1 | RMSE |
|---|---|---|---|---|---|---|---|---|
| Mean | 0.6696 | 0.6540 | 1.613 | 1.836 | 42.15 | 50.65 | 0.2521 | 0.3050 |
| Concept Bottleneck | 0.8298 | 0.7279 | 1.229 | *1.476* | 26.33 | 33.49 | 0.1474 | 0.1835 |
| Deep Model - Small | 0.4431 | 0.5006 | 1.238 | 1.637 | 30.72 | 37.03 | **0.0503** | **0.0727** |
| Deep Model - Large | *0.3974* | *0.4843* | *1.170* | 1.478 | **21.15** | **27.86** | 0.1487 | 0.1895 |
| Zero-shot | 0.4702 | 0.5371 | 1.525 | 1.754 | 38.80 | 46.41 | 0.2134 | 0.2610 |
| **Ours** | **0.2607** | **0.3778** | **1.077** | **1.314** | *24.79* | *32.99* | *0.0902* | *0.1260* |

### 4.2.3 Aboveground Biomass

**Observation Dataset:** Similar to poverty estimation, the observation variables are an input location and the output AGB estimate. We use NASA's GEDI (Dubayah et al., 2020) to obtain the observation value for three states in USA. We use data from the state of Massachusetts and Maine (North-East) as the train/test set and Washington (NorthWest) as the out-of-distibution set.

**Metric and Primitives:** We use L2 error as the metric and the same primitives as poverty estimation.

### 4.2.4 Contiguous Solar Induced Chlorophyll Fluorescence (CSIF)

**Observation Dataset:** The goal is to forecast CSIF values, by observing past CSIF and environmental values. Unlike other problems, which could make use of spatial information and thus required satellite images, CSIF forecasting is done without using satellite images. On the other hand, since this is a forecasting problem, is uses past CSIF values as well as past and current values of environmental variable. The observation variables are an input location and output CSIF values. The data comes from ERA5 climate data store (Hersbach et al., 2020).

**Metric and Primitives:** We use L2 error for each location as the evaluation metric. Along with the mathematical and logical operations. The primitive allows obtaining present environmental variables as well as a time series of past environmental variables such as minimum and maximum temperature of past months, or soil moisture index (see appendix B).

### 4.3 Experimental Setup

For the same set of training data we compare our best generated hypotheses with a set of baselines.

1. **Mean:** A naive baseline that use the mean of the training observation as the prediction.

2. **Concept Bottleneck (Linear):** Similar to Koh et al. (2020), we first extract a list of relevant features and train a linear classifier on it. This method is interpretable due to the bottleneck, however it is not very expressive (see appendix E.1).

3. **Deep models:** We use deep models such as Resnets and LSTM as baseline (see appendix E.2 for details). We use a small and large variant for each.

4. **Zero-shot:** This baseline tests how good would LLMs be on their own in generating hypotheses solely relying on prior knowledge without any observation. Since the generated programs can vary drastically, we report an average of 5 different zero-shot programs.

### 4.4 Results and Discussion

We first test our hypothesis on unseen but *in-domain* observations/regions close to the regions used for training (table 1). We observe that DiSciPLE outperforms all interpretable baselines. It can even outperform a deep model in many cases, specifically on population density estimation, while being significantly more interpretable. DiSciPLE also outperforms zero-shot program inference from LLMs suggesting that LLMs only have incomplete information about scientific domains and our evolutionary process is necessary to identify a better hypothesis.

Table 2: Performance of our hypotheses on out-of-distribution observations across various problems. This shows the reliability of programs produced by DiSciPLE (**red** is best and *blue* is second best).

| | Population Density | | Spatial Poverty | | AGB | | Temporal CSIF | |
| --- | --- | --- | --- | --- | --- | --- | --- | --- |
| | L2-Log | L1-Log | L1 | RMSE | L1 | RMSE | L1 | RMSE |
| Mean | 0.6734 | 0.6561 | 1.591 | 1.844 | 74.15 | 83.02 | 0.2393 | 0.3018 |
| Concept Bottleneck | 0.7951 | 0.7112 | 1.257 | 1.504 | 44.19 | 63.52 | 0.1565 | 0.2029 |
| Deep Model - Small | 0.6623 | 0.5967 | *1.284* | *1.654* | *35.27* | *53.06* | *0.1061* | *0.1614* |
| Deep Model - Large | *0.4460* | *0.5115* | 1.344 | 1.741 | 35.41 | 70.30 | 0.1815 | 0.2435 |
| Zero-shot | 0.7020 | 0.6412 | 1.510 | 1.773 | 55.11 | 64.32 | 0.2190 | 0.2814 |
| **Ours** | **0.3807** | **0.4426** | **1.134** | **1.420** | **31.10** | **42.93** | **0.0882** | **0.1256** |

**Are our hypotheses reliable?**  If a hypothesis is reliable it should be able to generalize to other regions.  table 2 shows DiSciPLE to these baselines on such a out-of-distribution set.  Here our approach outperforms all baselines *including deep networks*, suggesting that due to its interpretable-by-design representation our method learns model that can generalize better and overfit less.

**Are our hypotheses data-efficient?**  Our methods are only trained on a maximum of 4000 observations.  section 4.4 further shows that even when the amount of training data is reduced, our approach shows minimal degradation in performance compared to deep networks.  This suggests that while deep models can learn to generalize with a lot more data, our model does not need as much data to begin with, making it data-efficient.

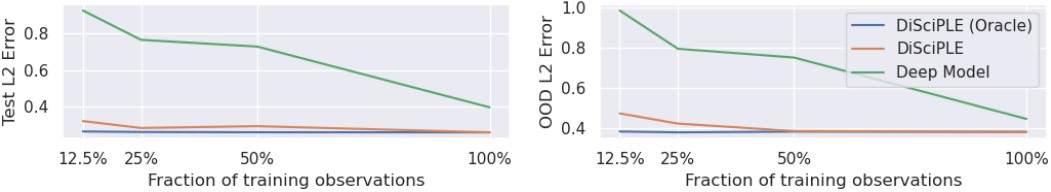

Figure 4: Performance of DiSciPLE compared to deep baselines as we reduce the amount of training observation (in terms of L2 error). The Oracle (blue) uses program learned from all observations, but uses only partial observation for parameter training. DiSciPLE (orange) uses partial observation during evolution as well. While the errors get worse as we reduce the observation data, the drop is significantly less severe for DiSciPLE compared to deep models, that tend to overfit.

**Are our hypotheses interpretable?**  Our hypotheses are interpretable-by-design as we can visualize the factors contributing to performance. Fig. 3 shows such programs (left in each card) for all four of our problems. An expert who is working with our method to figure out such hypotheses can add/edit parts of the formula and figure out which/how much do each of these components matter.

We perform this step of understanding the influence of individual operation by removing each operation in our program, and measure its affects on the final score. The DAGs on the right of each program show the program structure and the red edges show the influence of each component proportional to the width. This visualization can allow experts to understand which which operations are important for the model. For example, in the program graph for population density fig. 3, we can see that for semantic concepts such as "highway" and "residential building" are very important.

**Can our method perform better than expert humans?**  Our method would only be useful in real-world scenarios, if it can come up with stronger or comparable hypotheses to human experts. We test this on the task of AGB, by providing an expert (a PhD student actively working on AGB) with a user interface with the same information as our method. The experts took about 1.5 hours to use their domain knowledge and iterate over their program for AGB estimation. However, the best program they could come up with had an L1 error of **37.65** on the in-distribution set and **53.20** on the OOD set (compared to **24.79** and **31.10** for DiSciPLE). We figure this is primarily because

Table 3: Performance of our method as we successively remove the components. Both critic and simplification lead to performance improvement for our method.

| | | | Test | | OOD | |
|:---:|:---:|:---:|:---:|:---:|:---:|:---:|
| Feature-set pred. | Critic | Simplification | L2 log | L1 log | L2 log | L1 log |
| ✗ | ✗ | ✗ | 0.3159 | 0.4296 | 0.4835 | 0.5178 |
| ✓ | ✗ | ✗ | 0.2906 | 0.4049 | 0.4258 | 0.4826 |
| ✓ | ✓ | ✗ | 0.2873 | 0.3984 | 0.4184 | 0.4684 |
| ✓ | ✓ | ✓ | **0.2607** | **0.3778** | **0.3807** | **0.4426** |

experts need to spend more time on the problem. In general, experts would spend numerous days to come up with a good hypothesis, while our method can come up with a better hypothesis faster.

### 4.5 ABLATIONS

**How important is the role of feature-set prediction, critic, and simplification?**  Table 3 measures the performance of our model on the task of population density as we successively add these components to the evolutionary algorithm. The addition of feature set prediction instead of a single feature helps, as it allows our method to learn expressive linear regression parameters instead of letting the LLM come up with them. Further adding critic results further improvement as the programs start covering nicher concepts resulting in better unseen and OOD generalization. Finally adding in simplification also improves hypothesis. We posit that simplification removes irrelevant features preventing the LLM to focus on them when performing crossovers.

**How important are common sense and prior knowledge of LLMs?**  The two major differences or our method from symbolic regression are: 1) better crossover and mutation as LLMs can understand the meaning of the primitive functions. 2) use of prior knowledge for better guided search. Therefore we remove these two sources of information and test how well can our method perform. To remove the understanding of functions we rename them with meaningless terms and remove the descriptions. To remove the context of the problem we remove objective prompt. Without common sense, the search cannot even progress away from the initial random programs, resulting in worse-than-mean results (L1 error of 0.84 vs 0.26 for DiSciPLE on density estimation). This suggests that symbolic regression models, that have no understanding of open-world primitives, would struggle to search. If we just remove the context of the problem the model does slightly better and can obtain results better than the mean and zero-shot hypotheses (L1 error of 0.45). This suggests that while the search is moving in the objective's direction, it is slow. (see appendix D.3 for the complete table). We also present more ablations of our method in appendix D.

## 5  DISCUSSION AND CONCLUSION

**Limitations:**  A limitation of DiSciPLE is that we can only differentiably optimize learnable parameters in the last computational layer. This could miss out on hypotheses with useful parameters in some intermediate computation layers. We attempted to make the whole pipeline differentiable, however the model performance did not improve much. Many of the operations in our pipeline even though differentiable have zero-gradient in large part of input space, making gradient optimization challenging. Moreover, a completely differentiable hypothesis is even slower to optimize resulting in much slower evolution. In future work, we plan to use initialization tricks for non-linear optimization and second-order optimization to obtain even more expressive models.

**Conclusion:**  We present DiSciPLE – an evolutionary algorithm that leverages the prior-knowledge and common sense abilities of LLMs to create *interpretable, reliable and data-efficient* hypotheses for real-world scientific problem. This allows us to create hypotheses that are more powerful than existing interpretable counterparts and more insightful than deeper uninterpretable models while being on-par in terms of performance. We shows its prowess on 2 different expert scientific domain on 4 different problems. We believe that using DiSciPLE in tandem with a human expert can rapidly speed up the scientific process and result in numerous novel discoveries.

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

## A CROSSOVER AND MUTATION PROMPTS

For crossover we take two programs and their corresponding scores. We use the following prompt for crossover:

$\{\{h_1\}\}$

```
This program has a score of {{s(h₁)}}.
```

$\{\{h_2\}\}$

```
This program has a score of {{s(h₂)}}.

Can you write a function that gives higher score?  Feel free to
combine elements that worked from both programs.  Only give me
code.
```

And similarly we use the following prompt for random mutation.

$\{\{h\}\}$

```
This program has a score of {{s(h)}}.

Can you edit this code to write a better function for the problem?
Only give me code.
```

## B PROBLEM SPECIFIC PRIMITIVE DESCRIPTION

### B.1 PRIMITIVES AND THEIR DESCRIPTIONS FOR POPULATION DENSITY

```python
def elementwise_max(matrix1, matrix2):
    """
    Compute the element-wise maximum of two matrices.

    Parameters:
        matrix1 (numpy.ndarray): First input matrix.
        matrix2 (numpy.ndarray): Second input matrix.

    Returns:
        numpy.ndarray: Element-wise maximum of the input matrices.
    """

def elementwise_min(matrix1, matrix2):
    """
    Compute the element-wise minimum of two matrices.

    Parameters:
        matrix1 (numpy.ndarray): First input matrix.
        matrix2 (numpy.ndarray): Second input matrix.

    Returns:
        numpy.ndarray: Element-wise minimum of the input matrices.
    """

def elementwise_sum(matrix1, matrix2):
    """
    Compute the element-wise sum of two matrices.

    Parameters:
        matrix1 (numpy.ndarray): First input matrix.
        matrix2 (numpy.ndarray): Second input matrix.

    Returns:
```

```python
            numpy.ndarray: Element-wise sum of the input matrices.
        """

    def elementwise_product(matrix1, matrix2):
        """
        Compute the element-wise product of two matrices.

        Parameters:
            matrix1 (numpy.ndarray): First input matrix.
            matrix2 (numpy.ndarray): Second input matrix.

        Returns:
            numpy.ndarray: Element-wise product of the input matrices.
        """

    def elementwise_division(matrix1, matrix2):
        """
        Compute the element-wise division of two matrices.

        Parameters:
            matrix1 (numpy.ndarray): First input matrix.
            matrix2 (numpy.ndarray): Second input matrix.

        Returns:
            numpy.ndarray: Element-wise division of the input matrices.
        """

    def matrix_scalar_multiplication(matrix, scalar):
        """
        Perform matrix scalar multiplication.

        Parameters:
            matrix (numpy.ndarray): Input matrix.
            scalar (int or float): Scalar value.

        Returns:
            numpy.ndarray: Result of matrix scalar multiplication.
        """

    def elementwise_log(matrix):
        """
        Compute the element-wise logarithm of a matrix.

        Parameters:
            matrix (numpy.ndarray): Input matrix.

        Returns:
            numpy.ndarray: Element-wise logarithm of the input matrix.
        """

    def elementwise_exponentiate(matrix, base):
        """
        Compute the exponentiation of each element of a matrix with a given
        ↪  base.

        Parameters:
            matrix (numpy.ndarray): Input matrix.
            base (int or float): Base value.

        Returns:
            numpy.ndarray: Exponentiated matrix.
        """

    def min_pixel_distance_to_mask(mask):
```

```
        """
        Compute the minimum pixel distance from each pixel to a mask.

        Parameters:
            mask (numpy.ndarray): Binary mask array where 1 represents the
            ↪  mask and 0 represents the background.

        Returns:
            numpy.ndarray: Minimum pixel distance to the mask.
        """

def segment(im, text_prompt="trees"):
        """
        Segments a satellite image based on a text prompt. The text prompt
        ↪  can only take one concept at a time.

        Parameters:
            im (numpy.ndarray): An rgb satellite image.
            text_prompt (str): a text prompt

        Returns:
            mask (numpy.ndarray): Binary mask array where 1 represents the
            ↪  mask and 0 represents the background.
        Example:
        text_prompt can be but not limited to these:
        ["tennis", "skate park", "football field", "swimming pool",
        ↪  "cemetery", "multi-storey garage", "golf", "roundabout",
        ↪  "parking lot", "supermarket", "school", "marina", "baseball
        ↪  field", "fall", "pond", "airport", "beach", "bridge", "religious
        ↪  building", "residential building", "warehouse", "office
        ↪  building", "farmland", "university building", "forest", "lake",
        ↪  "nature reserve", "park", "sand", "soccer field", "equestrian
        ↪  club", "shooting range", "ice-rink", "commercial area",
        ↪  "garden", "dam", "railroad", "highway", "river", "wetland",
        ↪  "non-residential buildings", "coastline"]
        """
```

## B.2 PRIMITIVES AND THEIR DESCRIPTIONS FOR AGB AND POVERTY PREDICTION

The evolutionary search uses all the above defined functions, plus the following:

```
def get_satellite_image(location):
        """
        Get the satellite image for a given location.
        Parameters:
            location (tuple): Tuple containing the latitude and longitude of
            ↪  the location.
        Returns:
            numpy.ndarray: Satellite image for the location.

        Can be used for segmentation ONLY.
        """

def get_temperature(location):
        """
        Get the average annual temperature for a given location.
        Parameters:
            location (tuple): Tuple containing the latitude and longitude of
            ↪  the location.
        Returns:
```

```
            float: Temperature for the location normalized between 0 and
            ↪    255.
        """

    def get_precipitation(location):
        """
        Get the average annual precipitation for a given location.
        Parameters:
            location (tuple): Tuple containing the latitude and longitude of
            ↪    the location.
        Returns:
            float: Precipitation for the location between 0 and 255.
        """

    def get_elevation(location):
        """
        Get the elevation for a given location.
        Parameters:
            location (tuple): Tuple containing the latitude and longitude of
            ↪    the location.
        Returns:
            float: Digital Elevation for the location (scaled 0-8000) to
            ↪    0-255.
        """

    def get_nightlight_intensity(location):
        """
        Get the average annual nightlight intensity for a given location.
        Parameters:
            location (tuple): Tuple containing the latitude and longitude of
            ↪    the location.
        Returns:
            float: Nightlight intensity for the location (between 0 and 1).
        """

    def get_average(segmented_image):
        """
        Get the average pixel value of a segmented image.

        Parameters:
            segmented_image (numpy.ndarray): Segmented image.

        Returns:
            float: Average pixel value of the segmented image.
```

### B.3 PRIMITIVES AND THEIR DESCRIPTIONS FOR CSIF FORECASTING

For CSIF forecast, the API borrows mathematical and logical functions from above. Additionally it has the following to obtain more environmental variables.

```
    def get_historical_csif(locations, num_months=36):
        """
        Get historical CSIF (contiguous solar induced chlorophyll
        ↪    fluorescence) time-series data for the last given number of
        ↪    months.

        Parameters:
            locations: a list of locations in a specific format.
            num_months (int): Last number of months to get time-series data
            ↪    for.
```

```python
 10      Returns:
 11          numpy.ndarray: Historical time series data for the given
             ↪  locations. size = (len(locations), num_months)
 12      """
 13
 14  def get_historical_min_temperature(locations, num_months=36):
 15      """
 16      Get historical minimum temperature time-series data for the last
         ↪  given number of months.
 17
 18      Parameters:
 19          locations: a list of locations in a specific format.
 20          num_months (int): Last number of months to get time-series data
             ↪  for.
 21      Returns:
 22          numpy.ndarray: Historical time series data for the given
             ↪  locations. size = (len(locations), num_months)
 23      """
 24
 25
 26  def get_historical_max_temperature(locations, num_months=36):
 27      """
 28      Get historical maximum temperature time-series data for the last
         ↪  given number of months.
 29
 30      Parameters:
 31          locations: a list of locations in a specific format.
 32          num_months (int): Last number of months to get time-series data
             ↪  for.
 33      Returns:
 34          numpy.ndarray: Historical time series data for the given
             ↪  locations. size = (len(locations), num_months)
 35      """
 36
 37
 38  def get_historical_radiation(locations, num_months=36):
 39      """
 40      Get historical solar radiation time-series data for the last given
         ↪  number of months.
 41
 42      Parameters:
 43          locations: a list of locations in a specific format.
 44          num_months (int): Last number of months to get time-series data
             ↪  for.
 45      Returns:
 46          numpy.ndarray: Historical time series data for the given
             ↪  locations. size = (len(locations), num_months)
 47      """
 48
 49  def get_historical_precipitation(locations, num_months=36):
 50      """
 51      Get historical precipitation time-series data for the last given
         ↪  number of months.
 52
 53      Parameters:
 54          locations: a list of locations in a specific format.
 55          num_months (int): Last number of months to get time-series data
             ↪  for.
 56      Returns:
 57          numpy.ndarray: Historical time series data for the given
             ↪  locations. size = (len(locations), num_months)
 58      """
 59
 60
```

```python
def get_historical_photoperiod(locations, num_months=36):
    """
    Get historical photoperiod time-series data for the last given
    ↪  number of months.

    Parameters:
        locations: a list of locations in a specific format.
        num_months (int): Last number of months to get time-series data
        ↪  for.
    Returns:
        numpy.ndarray: Historical time series data for the given
        ↪  locations. size = (len(locations), num_months)
    """

def get_historical_swvl1(locations, num_months=36):
    """
    Get historical soil water content in the first layer time-series
    ↪  data for the last given number of months.

    Parameters:
        locations: a list of locations in a specific format.
        num_months (int): Last number of months to get time-series data
        ↪  for.
    Returns:
        numpy.ndarray: Historical time series data for the given
        ↪  locations. size = (len(locations), num_months)
    """

def get_present_min_temperature(locations):
    """
    Get present minimum temperature data for the given locations.

    Parameters:
        locations: a list of locations in a specific format.
    Returns:
        numpy.ndarray: Present minimum temperature data for the given
        ↪  locations. size = (len(locations),)
    """

def get_present_max_temperature(locations):
    """
    Get present maximum temperature data for the given locations.

    Parameters:
        locations: a list of locations in a specific format.
    Returns:
        numpy.ndarray: Present maximum temperature data for the given
        ↪  locations. size = (len(locations),)
    """

def get_present_radiation(locations):
    """
    Get present solar radiation data for the given locations.

    Parameters:
        locations: a list of locations in a specific format.
    Returns:
        numpy.ndarray: Present solar radiation data for the given
        ↪  locations. size = (len(locations),)
    """

def get_present_precipitation(locations):
```

```
        """
        Get present precipitation data for the given locations.

        Parameters:
            locations: a list of locations in a specific format.
        Returns:
            numpy.ndarray: Present precipitation data for the given
            ↪  locations. size = (len(locations),)
        """

def get_present_photoperiod(locations):
        """
        Get present photoperiod data for the given locations.

        Parameters:
            locations: a list of locations in a specific format.
        Returns:
            numpy.ndarray: Present photoperiod data for the given locations.
            ↪  size = (len(locations),)
        """

def get_present_swvl1(locations):
        """
        Get present soil water content in the first layer data for the given
        ↪  locations.

        Parameters:
            locations: a list of locations in a specific format.
        Returns:
            numpy.ndarray: Present soil water content in the first layer
            ↪  data for the given locations. size = (len(locations),)
        """
```

## C MORE EXAMPLES

### C.1 MORE EXAMPLES OF CROSSOVER

**Parent 1**

```
def estimator(location):
    images = get_satellite_image(location)
    temperature = get_temperature(location)
    precipitation = get_precipitation(location)
    nightlight = get_nightlight_intensity(location)
    return temperature, precipitation, elevation, nightlight
```

**Parent 2**

```
def estimator(location):
    images = get_satellite_image(location)
    roads = segment(images, 'roads')
    avg_roads = get_average(roads)
    avg_nightlight = get_nightlight_intensity(location)
```

**Program after crossover**

```python
def estimator(location):
    images = get_satellite_image(location)
    roads = segment(images, 'roads')
    avg_roads = get_average(roads)
    avg_nightlight = get_nightlight_intensity(location)
    avg_precipitation = get_precipitation(location)
    avg_temperature = get_temperature(location)
    return (avg_roads * avg_precipitation * avg_nightlight *
        ↪  avg_temperature)
```

**Parent 1**

```python
def estimator(location):
    images = get_satellite_image(location)
    forests = segment(images, 'forests')
    avg_forests = get_average(forests)
    roads = segment(images, 'roads')
    avg_roads = get_average(roads)
    elevation = get_elevation(location)
    nightlight_intensity = get_nightlight_intensity(location)
    return avg_forests, avg_roads, 1 / (1 + elevation), 1 / (1 +
        ↪  nightlight_intensity)
```

**Parent 2**

```python
def estimator(location):
    images = get_satellite_image(location)
    poverty_mask = segment(images, 'poverty')
    return get_average(poverty_mask)
```

**Program after crossover**

```python
def estimator(location):
    images = get_satellite_image(location)
    poverty_mask = segment(images, 'poverty')
    education_mask = segment(images, 'education')
    health_mask = segment(images, 'health')
    avg_poverty = get_average(poverty_mask)
    avg_education = get_average(education_mask)
    avg_health = get_average(health_mask)
    elevation = get_elevation(location)
    nightlight_intensity = get_nightlight_intensity(location)
    return avg_poverty, avg_education, avg_health, 1 / (1 + elevation),
        ↪  1 / (1 + nightlight_intensity)
```

### C.2 ILLUSTRATION OF SIMPLIFCATION

For the following program, we show the steps of simplification. **Program after crossover**

```python
def estimator(im):
    building_mask = segment(im, "residential building")
    nr_mask = segment(im, "non-residential buildings")
    vegetation_mask = segment(im, "forest")
    water_mask = segment(im, "lake")
    road_mask = segment(im, "highway")
```

Table 4: different LLMs

| LLM | Test | | OOD | |
|---|---|---|---|---|
| | L2 log | L1 log | L2 log | L1 log |
| *Qwen2.5-7b-instruct* | 0.2950 | 0.4039 | 0.4263 | 0.4716 |
| *llama-3-8b-instruct* | 0.2626 | 0.3842 | 0.3946 | 0.4514 |
| *llama-3.1-8b-instruct* | 0.2771 | 0.3872 | 0.3950 | 0.4518 |
| *llama-3.1-70b-instruct* | 0.2896 | 0.3958 | 0.4223 | 0.4663 |
| Deep Model - Large | 0.3974 | 0.4843 | 0.4460 | 0.5115 |

```
7
8    building_distance = min_pixel_distance_to_mask(building_mask)
9    nr_distance = min_pixel_distance_to_mask(nr_mask)
10   vegetation_distance = min_pixel_distance_to_mask(vegetation_mask)
11   water_distance = min_pixel_distance_to_mask(water_mask)
12   road_distance = min_pixel_distance_to_mask(road_mask)
13
14   return building_distance, nr_distance, vegetation_distance,
       ↪   road_distance
```

In the first step, as can be seen in fig. 5 (top-left graph), at the bottom right there is a leaf node that is not a return node. We remove that node and other leaf nodes recursively resulting in a graph like top-right.

Using regression weights our method figures out that the left most branch or `building_distance` is not a useful value to be returned. So in the third step we remove that and recursively all the leaf nodes.

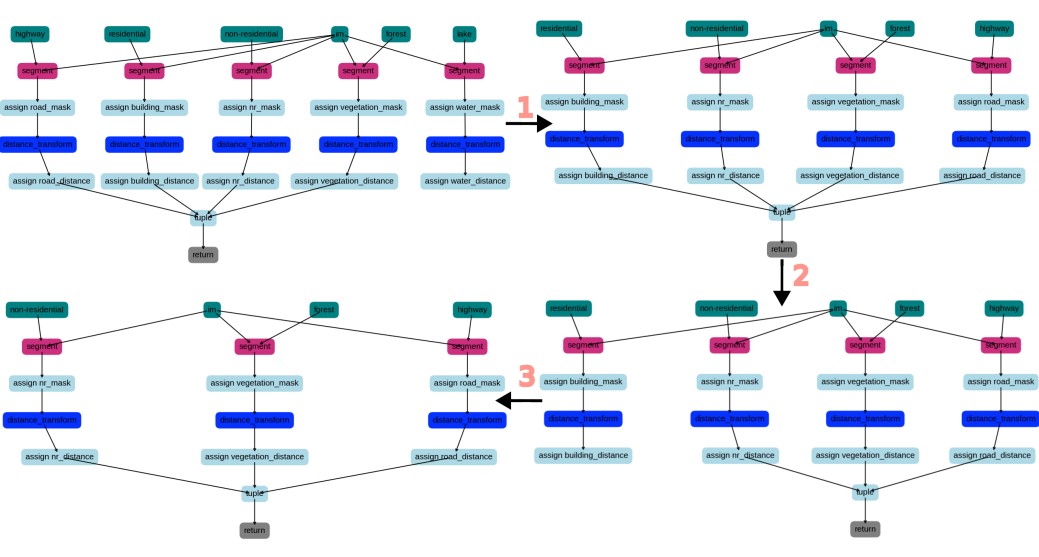

Figure 5: Process of simplification illustrated over a function.

## D ADDITIONAL ABLATIONS

### D.1 USING DIFFERENT LLMS

A key contribution of our work is to leverage the common-sense knowledge in LLMs to improve evolutionary search. So, it is natural to question whether (a) LLMs (with similar capacity) trained with a different large corpus of text would generate hypotheses with different levels of reliability and

Table 5: Perfomance of our method when removing the context of the problem (objective prompt from the evolution, and when renaming and not describing the primitive functions to the LLM. We see significant drops in performance in both cases, suggesting that both the common-sense and prior knowledge of LLM is important to perform efficient evolutionary search. )

| Method | L1 log | L2 log |
|---|---|---|
| No common-sense | 0.8401 | 0.7186 |
| No problem context | 0.4498 | 0.5140 |
| DiSciPLE full | **0.2607** | **0.3778** |

(b) LLMs with larger capacity would produce more reliable hypotheses. We answer these questions by testing recent LLMs: Qwen-2.5/7b (Team, 2024), llama-3/8b, llama-3.1/8b, llama-3.1/70b. For ease of experimentation, we reduce the number of generations to 10 and the population size to 60. We report the results in table 4. DiSciPLE works robustly with various LLMs and could generate more reliable models/hypotheses than the Deep Model. While the performance of the hypotheses varies, we do not observe any discernible difference among the various hypotheses.

### D.2 ABLATION ON NOISY/UNRELIABLE PRIMITIVES

To investigate how accurate/robust should the underlying black-box model be?, we corroded the OSM maps with a 3x3 convolution and ran DiSciPLE to generate a new hypothesis. With the corroded OSM maps, we observe an L2 log error of 0.3713 on the test set — a large degradation in performance compared to clean OSM maps (L2 log error: 0.2626).

### D.3 ABLATIONS OF LLM COMMON-SENSE AND PRIOR KNOWLEDGE

table 5 shows more extensive evaluation performing evaluation without primitive understanding and problem understanding.

## E EXPERIMENTAL SETUP

### E.1 MORE DETAILS ON CONCEPT BOTTLENECK BASELINES

For the CSIF task, the concept bottleneck features are average of past CSIF and environmental variable and the current environmental variable. For all the other tasks, the bottleneck features are 42 categories of segments obtained from either OSM or GRAFT and the environmental variable.

### E.2 MORE DETAILS ON DEEP MODELS BASELINE

For spatial tasks, we use a ResNet-18 (He et al., 2016) based U-Net (Ronneberger et al., 2015) (Deep Model-Large). Since our model also needs to be data-efficient, to prevent overfitting, we also try a smaller backbone of 4-layer fully convolutional network(Deep Model-Small). For CSIF, we use LSTMs Hochreiter & Schmidhuber (1997) of different depth as our big (12-layers) and small (4) models

## F MORE DETAILS

### F.1 LIST OF 42 LAND-USE CONCEPTS

Table 6 show the list of 42 concepts extracted from OpenStreetMaps and also used in GRAFT to get partitions for critic.

| | | | | | |
|---|---|---|---|---|---|
| tennis courts | skate park | american football field | swimming pool | cemetery | pond |
| golf course | roundabout | parking lot | supermarket | school | marina |
| baseball | waterfall | multi-storey parking garage | airport | beach | bridge |
| religious building | residential building | university building | office | farmland | warehouse |
| forest | lake | nature reserve | park | sandy area | soccer field |
| equestrian center | shooting range | non residential buildings | commercial area | garden | dam |
| railroad | highway | river or stream | wetland | ice-rink | coastline |

Table 6: List of concepts extracted from OSM and also used via GRAFT for critic data stratification.

