# OpenReview forum: "DiSciPLE: Learning Interpretable Programs for Scientific Discovery"
_ICLR.cc/2025/Conference — ICLR 2025 Conference Withdrawn Submission_

### Official Review · Reviewer_KhSF · 2024-10-31

**Soundness:** 1
**Presentation:** 3
**Contribution:** 1
**Rating:** 3
**Confidence:** 4

**Summary:**

The manuscript deals with the research question: How can we automatically discover expressive, interpretable and reliable hypotheses in a sample-efficient way? In order to do that, the propose a DiSciPLE framework plus a critic, and demonstrate its performance for a few chosen scientific domains.

**Strengths:**

First, it is noteworthy that it does not constrain the domain to certain scientific fields. This is especially ambitious because one would imagine the vast majority of hypotheses to look quite different in different fields, so a one-size-fit-all approach such as the one explored here is in my opinion best suited as a proof-of-feasibility, rather than a useable framework which would need to be more customized to the characteristics of different research domains.

The proposed DiSciPLE framework (+ the critic) is a evolutionary algorithm working on a set of lower-level primitives, for the most part to reduce the search space to something manageable. It should be noted here that this is in fact a greedy approach which can indeed identify interesting hypotheses given data, but which cannot claim to be able to identify all possible, or even the best hypotheses. The degree of the greediness should probably be an important hyperparameter, and depending on the choice the usecase for the framework would shift, from exploration-spotlight towards a more systematic sieving.

**Weaknesses:**

In its provided form, I think publication and broader dissemination of this framework is unfortunately premature. There are a number of critical pitfalls that are not sufficiently addressed as of yet:

- Identifying good primitives seems to be a critical aspect that may vary strongly from problem to problem (from dataset to dataset). The authors simply write "the experts (...) provide our framework with a set of primitive variables and functions", which is first strongly disincentivizing common usage of this framework (since it still requires significant customized effort, which the framework+critic are supposed to solve), and it also strongly constrains the search space to only work in a direction that has been previously identified by the experts. However, if the experts already know that much about the hypothesis -- why use this new framework in the first place?

This concern could be addressed by the authors systematically making a case that the primitive functions, which "change depending on the application domain", are in fact mostly general and maintained within said application domain, and can conceivably be provided by the framework itself (maybe as an additional module). This case is unfortunately not made.

The authors implement their system using only llama-3-8b-instruct. It would be necessary in my opinion to also try to replicate this result at least on one other open-source LLM model to evaluate how much of the results in such a highly specific prompting scheme is model-dependent.

Lastly, this manuscript does not address at all the problem of multiple hypothesis testing. By looking for any kind of pattern in the data fitting with the given primitives, it is highly likely that some of the "data efficient" hypotheses may in fact be spurious. A kind of correction for that, an LLM-analogue of Bonferroni correction or similar would urgently be needed to have more reliable potential hypotheses as outputs.

Overall, based on the above considerations and seeing how they are not addressed at all in the manuscript, I unfortunately need to recommend rejection for this manuscript.

**Questions:**

Based on the weaknesses as listed above:

Can you make a case that primitive functions are generalisable across many problems within a research field?

How do you address that this in effect leads to multiple hypothesis testing, without any correction? Do you see this differently, and if so, in what way?

Would you consider customizing your framework to a specific domain, then more thoroughly demonstrate its usefulness within said domain?

---

### Official Review · Reviewer_SCGV · 2024-11-02

**Soundness:** 3
**Presentation:** 2
**Contribution:** 3
**Rating:** 6
**Confidence:** 3

**Summary:**

The authors introduce a method to automatically produce scientific programs that perform operations (specified via an API of primitive functions) on raw input data/covariates. In brief, their method involves using an evolutionary algorithm, where the crossover and mutation operators are parameterized with LLMs, to construct “programs” that essentially compute lists of features; these programs can call library functions. These features are interpretable and can be used in a predictive model. The authors demonstrate reasonable performance in a number of tasks.

**Strengths:**

The problem setting is important, and the approach is overall practical and well-motivated. Given LLM’s domain knowledge, I think it makes a lot of sense to use LLMs (equipped with access to tools) to construct features for a scientific modeling task, and doing so in an iterative way makes a lot of sense.

The experimental results are pretty thorough and extensive. DiSCiPLE shows improvements over both naive and deep baselines, both in-domain and out-of-domain. DiSCiPLE is also data-efficient compared to deep models, requiring far fewer training observations. The ablations are comprehensive (e.g., removing domain knowledge to assess the role of the prior knowledge of LLM, characterizing the role of critic/simplification in overall performance). I appreciate that the authors used real-world datasets and it was nice to see a comparison with human experts.

**Weaknesses:**

The clarity of this paper can be improved (specifically the description of the method and details about the experiments).

I was confused about the relationship between the notion of a hypothesis and a scientific program in Section 3.1. The scientific programs, if I understand correctly, are essentially extracting lists of features that can be used to produce a prediction via either a regression model or direct prompting of an LLM. This is explained in Section 3.3 but needs to be clear in Section 3.1 since the targets and the hypothesis h(x_i) are introduced there.

I think the introduction added to this confusion because I thought the programs themselves were parameterizing hypotheses. However, to be precise, it’s actually the combination of these programs + LLM/linear model, right? Basically, this is a paper about feature selection for simple (linear?) predictive scientific models but that really wasn’t that clear to me until reading the paper a few times.

I was also a bit confused about how the thickness of the edges is computed in the DAG representation of the program.

I think a few key experimental details are missing from the main text. I think it’s important to be precise about what “small” and “large” mean for a deep model. How large are these datasets? How were these datasets chosen? Is there a reason you can’t compare to a symbolic regression baseline?


I was also confused about the motivation behind certain steps. Why not use Lasso or Ridge to control for model complexity in the program simplification step? The simplification step feels a bit ad-hoc and potentially unnecessary, given that there are well-established techniques for regularization. I could be mistaken though and am open to discussion on this point!

The tasks mainly focus on regression but I think scientific modeling can be more broad than that. I think this is reasonable as a starting point.

**Questions:**

The related work section can be expanded. For example, [1] is a relevant paper that uses LLMs to propose and critique probabilistic programs and it could be good to discuss the relationship/novelty of this work with respect to this previous work.

[1] Automated Statistical Model Discovery with Language Models. M. Y. Li, E. B. Fox, & N. D. Goodman. (2024). In International Conference on Machine Learning (ICML).

What does AGB estimation and CSIF forecasting mean? Do you define those somewhere?

Who determines the API of primitive functions? It seems a bit restrictive to require a domain expert to specify those in advance although I appreciate that this is probably a reasonable design choice in practice.

In the critic step, how are the categories for partitioning the data determined?

I’m a bit surprised that the deep models are so ineffective at this task. Did you try transformers or ConvNets for this task?

How does the performance vary as a function of the number of steps of the evolutionary algorithm?

---

### Official Review · Reviewer_Qcgr · 2024-11-02

**Soundness:** 2
**Presentation:** 3
**Contribution:** 2
**Rating:** 3
**Confidence:** 4

**Summary:**

The paper presents DiSciPLE, which combines LLMs with evolutionary search to generate interpretable scientific hypotheses as Python programs. The approach is evaluated on four datasets across two domains, showing better results compared to baselines in the paper.

**Strengths:**

The idea of LLM-driven knowledge-guided program search for interpretable scientific modeling is well-motivated and reasonable

**Weaknesses:**

* The paper's heavy reliance on manually designed primitives raises concerns about the LLM role in discovery.  Table 5 shows significant performance drops without these primitives on population density problem (L1 error: 0.26 to 0.84, L2 error: 0.37 to 0.71). Comparing results of "zero-shot" and "no common sense" variants in Table 1 and Table 5 also further supports this observation that primitives seem to contribute more to performance than the LLM-based evolutionary search for discovery. I understand that domain-specific context is necessary for LLMs, however, requiring so many manual primitives undermines the claims. Have authors explored using LLMs for the design of these problem-specific primitives?

* Following on that, the framework's use of only T=15 evolutionary iterations (line 322) raises concerns about whether the good performance stems from LLM-based search/discovery or simply from LLMs retrieving memorized knowledge and recombining manually pre-defined primitives.

* Can you provide analysis of program diversity across multiple runs/replications of framework for a fixed dataset?

* There are only three baselines in this study (concept bottleneck [1] for interpretable modeling) and two deep models (ResNet and LSTM) for comparison. While the paper demonstrates improvements over these baselines, this is insufficient for a thorough evaluation. I understand that the focus of this work is mainly on interpretable modeling so better deep baselines like transformers may not be the focus of this study. But [2] has been introduced in 2020. Comparison with more recent relevant baselines such as [2] or LLM-based ones such as [3] is essential for validating the method's effectiveness.

* A very similar idea has been recently explored in [4]. I wonder how the proposed method is different. Both approaches seem to leverage LLMs' prior knowledge and Python generation capabilities for program synthesis in scientific discovery.

* Table 4 shows larger Llama3 models performing worse than smaller ones. I would suggest authors to investigate if this is due to search randomness and consider reporting results across multiple runs.


**Minor Comments:**
* Definition of elevation feature in Figure 1 seems missing
* The paper doesn't clearly explain how problem-specific primitives are incorporated into LLM prompts during evolution.
* It's unclear whether Table 5 shows in-domain or OOD performance
* I suggest authors to provide examples of prompt modifications for ablation variants, particularly the ablations for common sense and problem context

---
[1] Koh et al., Concept Bottleneck Models, 2020

[2] Oikarinen et al., Label-Free Concept Bottleneck Models, 2023

[3]  Yang et al., Language in a Bottle: Language Model Guided Concept Bottlenecks for Interpretable Image Classification, 2023

[4] Shojaee et al., LLM-SR: Scientific Equation Discovery via Programming with Large Language Models, 2024

**Questions:**

check the weaknesses section

---

### Official Review · Reviewer_gv6g · 2024-11-03

**Soundness:** 2
**Presentation:** 3
**Contribution:** 3
**Rating:** 5
**Confidence:** 3

**Summary:**

The paper introduces DiSciPLE, a framework that combines Large Language Models (LLMs) with evolutionary search to generate interpretable scientific hypotheses as Python programs. The key idea is to leverage LLMs' prior knowledge and reasoning capabilities to guide the search process through meaningful mutations and crossovers, while maintaining interpretability through feature selection and linear regression. The framework includes two novel components: a program critic that evaluates hypotheses on stratified data partitions, and a program simplifier that removes redundant features. The authors evaluate their approach on four real-world tasks across two scientific domains (demography and climate science).

**Strengths:**

* Interesting combination of LLM reasoning with API specifications for visual understanding and iterative refinement based on observations
* Well-written paper with clear organization and easy-to-follow methodology
* Ablation studies demonstrating the contribution of different components

**Weaknesses:**

* Limited scope: The benchmark problems are primarily focused on satellite image analysis and don't fully support the claim of general scientific discovery
* Inadequate coverage of relevant prior work: Several recent studies using LLMs for symbolic regression, scientific hypothesis generation, and interpretable programming with similar approaches are not discussed or compared against
* Evaluation fairness concerns: The baselines aren't provided with the same processed features available to DiSciPLE through its primitives
* Limited baselines: Only uses concept bottleneck models while ignoring more recent approaches in interpretable ML.

**Questions:**

* The provided primitives contain significant domain knowledge (e.g., average temperature). While LLM can select among these primitives and simplification can remove redundant features, one might argue that why aren't other baselines (deep learning models, or potentially decision trees) given access to these processed features?

* Could the authors clarify the input features provided to the concept bottleneck and deep model baselines mentioned in Appendix E1?

* Why report the average of 5 zero-shot programs when DiSciPLE generates and selects from 100 programs per iteration? A fairer comparison might be generating 1500 programs (matching DiSciPLE's total of 15*100) without observation feedback and selecting the best.

* The number of iterations (10-15) seems low for effective discovery. Could the authors show convergence plots and compare with larger one-shot program generation?

* Several claims about novelty need revision, as LLMs for mutation/crossover operations have been studied in various domains (neural architecture search, interpretable programming, algorithm discovery, and symbolic regression). Could the authors clarify their specific contributions?

* How can the program critic approach, which works well for land-use categories, be extended to general scientific discovery problems?

* Are the data observations for different input features normalized? Simplification based on weights of linear regression would assume that the range of values are normalized, otherwise, small weights might contribute significantly to the output.

* The final program generates several features by selecting and having products of different raw features, and then use a linear addition (linear regression) of them. However, in many scientific domains, nonlinearities do exist. Can you comment on how to improve the framework for such cases?

* Why does performance drop with more powerful LLMs (LLaMA-70B vs 8B)? Why weren't more recent models like GPT-4 evaluated?

* How are parents sampled in Algorithm 1? Is there an ablation studying the relative importance of mutation vs crossover operators?

* a) In Figure 3's poverty example, using "poverty_mask" as a primitive seems problematic as it is the same as the output. b) Is the 0.01 coefficient in the AGB example from LLM knowledge? Does that have specific meaning? Can such parameters be optimized by the model?

Some relevant reference on LLM for scientific hypothesis discovery:
* LLM and Simulation as Bilevel Optimizers: A New Paradigm to Advance Physical Scientific Discovery
* LLM-SR: Scientific Equation Discovery via Programming with Large Language Models
* In-Context Symbolic Regression: Leveraging Large Language Models for Function Discovery
* Automated Statistical Model Discovery with Language Models
* Discovery Bench: Towards Data-Driven Discovery with Large Language Models

---

### Author Response · Authors · 2024-11-15
**Response to the reviewers.**

We thank the reviewers for their constructive feedback on our work. Based on the recommendations, we plan to withdraw this submission and resubmit a refined version primarily focusing on visual data. In the meantime, we address some of the major concerns below.

**Difference from Symbolic Regression (SR):** The scope of our paper is problems on high-dimensional real-world visual scientific problems (instead of lower-dimensional problems such as equation discovery for the Feynman formula). The approaches such as LLM-based variants of symbolic regression [1,2] do not work under such high-dimensional and open-world settings. As a result, they are always tested on problems with simpler mathematical primitives that are not open-world or very expressive.

Instead, we propose an approach that is **complementary** to these methods. Methods for SR cannot be applied directly in higher-dimensional open-world visual problems.  On the other hand, on low-dimensional problems, existing tools for SR such as [1,2] would perform better than our method. We will make this distinction clear, by focusing our paper on high-dimensional visual scientific problems.

We would also like to point out that many of the LLM-assisted SR works [1,2] are parallel submissions. Therefore, we believe our work should not be penalized regarding methodological novelty with respect to them.

**Reliance on primitives:** Adding on to the previous point, to build an interpretable yet powerful hypothesis for high-dimensional scientific problems, powerful domain-specific primitives are needed. The lack of such primitives is the primary reason why prior work on symbolic regression cannot work well on high-dimensional problems.

We would also like to clarify that such primitives are relatively easy to obtain in different modalities. First, the mathematical and logical primitives are domain-independent and should be used for all problem domains. Second, experts in a scientific domain have good knowledge of the domain-specific environment variables, therefore all such environment variables can be easily passed. Finally, several domains have vision foundation models that again can be easily used as primitives.

To conclude, adding a large number of domain-specific primitives is significantly less work for experts compared to iterating over the hypothesis. Moreover, access to such open-world primitives is what makes DiSciPLE work on high-dimensional data.

**Evaluation information to baseline:** We provide the same input to all the baselines that can be provided to our method. For example, if the programs are allowed to use environment variables, we also allow our concept bottleneck model and deep model with those environment variables.

**Strength of the baselines**: Not only do the baselines see the same information, but the model architecture for deep baselines is also based on state-of-the-art works (metzger et. al., nathaniel et. al.) in the areas of population density, poverty, AGB or CSIF. We will make this information clear in the paper.

We promise to add more experiments to address the remaining concerns in the future iteration of our work.

Thank again you for your reviews.

[1] Grayeli et al., Symbolic Regression with a Learned Concept Library, 2024

[2] Shojaee et al., LLM-SR: Scientific Equation Discovery via Programming with Large Language Models, 2024

---

### Note · Authors · 2024-11-15

I have read and agree with the venue's withdrawal policy on behalf of myself and my co-authors.